# Study on the Design and Cutting Performance of a Revolving Cycloid Milling Cutter

**Guangyue Wang, Xianli Liu, Weijie Gao, Bingxin Yan and Tao Chen \***

School of Mechanical and Power Engineering, Harbin University of Science and Technology, Harbin 150080, China

**\*** Correspondence: chentao@hrbust.edu.cn; Tel.: +86-180-4517-3177

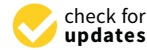

**Featured Application: This research is beneficial to industrial applications. The likely users could be titanium alloy products manufacturers.**

**Abstract:** Problems such as low machining efficiency, severe tool wear and difficulty in safeguarding surface quality always exist in the machining process of titanium alloy with ball-end milling cutters. To address these issues, the design and manufacture of a revolving cycloid milling cutter for titanium alloy processing were studied in this paper. Firstly, the mathematical model of the revolving cycloid milling cutter contour surface was established. The parametric equation of an orthogonal helix cutting edge curve of a revolving cycloid milling cutter is presented. Then, the bottom boundary curve of the rake face is introduced. The five-axis grinding trajectory equation of revolving cycloid milling cutter rake face was derived based on the edge curve equation and coordinate transformation. Next, fabricating the revolving cycloid milling cutter and detecting the grinding accuracy of tool profile and geometric angle were performed. At last, a contrast test regarding the performance of the revolving cycloid milling cutter and the ball-end milling cutter in cutting titanium alloy TC11 was carried out. According to the test results, in comparison to the ball-end milling cutter, the revolving cycloid milling cutter had a smaller ratio of the axial force to the tangential force. Moreover, its flank face wore more slowly and evenly. As a result, a good surface processing quality can be maintained even under larger wear conditions, demonstrating an outstanding cutting performance.

**Keywords:** revolving cycloid milling cutter; cutting tool design; tool fabrication; cutting performance; titanium alloy

## 1. Introduction

Featuring high specific strength, high creep corrosion resistance, and high wear resistance, titanium alloy has been extensively used in aviation, aerospace, energy and biomedical fields [1]. However, the physical properties like low thermal conductivity, small elastic modulus and large friction coefficient of titanium alloy may lead to small tool-chip contact area, large unit cutting force, and high cutting temperature in the cutting process [2–6], seriously restricting the machining quality and application of titanium alloy parts. In the meantime, the titanium alloy workpiece surface is complex and diverse, but the variety of cutting tools is single, which leads to the unreasonable application of cutting tools and prominent failure problem. In order to improve processing efficiency and reduce manufacturing costs, it is imperative to develop new cutting tools.

The design and fabrication of new tools have been studied by scholars at home and abroad. Liu et al. [7] studied the optimization of the edge shape optimization of the radius-end milling cutter to greatly enhance the machining efficiency of the radius-end milling cutter. Jin et al. Reference [8] designed two ball-end milling cutters for molds machining through optimizing the cutting edge

and the rake angle of the ball-end milling cutter. In this way, the contour accuracy of the tool and the anti-damage performance of the cutting edge can be well improved. Chen et al. [9] carried out structural design and grinding of gradient chamfering cutters. They put forward the developed tool grinding accuracy detection method and validated the cutting performance of the developed tool. Pham et al. [10] proposed a five-axis grinding manufacturing model for the face milling cutter. Nguyen et al. [11] established the mathematical model of cutting edge line of ball-end milling cutter. Based on the contact relationship between the grinding wheel and the rake face, the grinding trajectory equation of grinding wheel was derived. The validation of the model was verified by grinding experiments. Tan et al. [12] obtained the influence of the milling method on the cutting force, tool wear and surface integrity by carrying out the experimental study of inclination milling of titanium alloy with ball-end mill. It was found that the cutting force generated by upward milling along the angle was larger, and the surface quality was the best. The tool life is the longest when milling downward along the angle. Krishnaraj et al. [13] detected that the cutting depth has the maximum influence on the cutting force, while the cutting speed has a significant influence on temperature, after conducting a study on the high-speed cutting TC4 test of the carbide-tipped milling cutter.

These scholars have done a great deal of excellent work in the design and manufacturing of new cutting tools. The existing research focuses on helical edge design, geometric angle optimization and grinding manufacturing of ball-end milling cutters [14–19]. Ball-end milling cutters have been widely used in the field of machining, and because of the limitation of processing equipment and grinding method, there has been little research on new cutting tools for titanium alloy processing. Therefore, aiming at the problems of serious tool wear, low efficiency and quality in milling titanium alloy, the design and manufacture of revolving cycloid milling cutter and its cutting performance have been studied in this paper.

## 2. Design of the Revolving Cycloid Milling Cutter

A revolving cycloid milling cutter was proposed to enhance the quality and efficiency of processing titanium alloy parts. Mathematical models of the contour surface of revolving cycloid milling cutter and the orthogonal helix cutting edge have been established. By numerical simulation, the effective cutting helix angle and the scallop height of the machined surface of a revolving cycloid milling cutter and a ball-end milling cutter are compared.

### 2.1. Establishment and Simulation of Geometric Model for the Revolving Cycloid Milling Cutter

Figure 1a is the forming principle of a cycloid. When a cycle with the diameter of 2*a* rolls along the *X*-axis in the XOZ coordinate system, the trajectory formed by any selected point on the circumference is called a cycloid. Cycloids are widely used in industry, and their width and height correspond to each other.

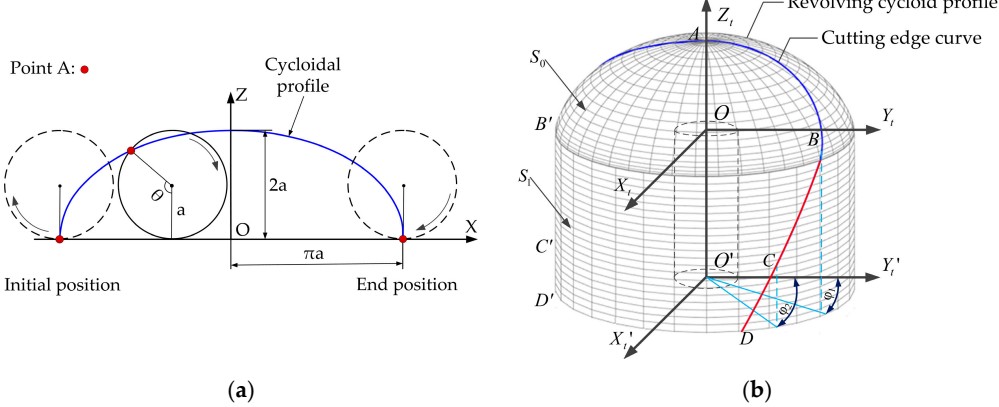

**Figure 1.** Geometric model of a revolving cycloid milling cutter: (**a**) The forming principle of a cycloid; (**b**) contour surface and cutting edge curve of a revolving cycloid milling cutter.

The coordinate of the circle center is set as point $(-\pi a, a)$, point A coinciding with the $X$-axis at $(-\pi a, 0)$ is defined as the initial position of the circle. After the circle rolls clockwise along the $X$-axis in one round, the position where point A coincides again with the $X$-axis and is defined as the end position of the circle. On this basis, the trajectory of point A can be expressed as:

$$\begin{cases} x = a(\theta - \sin\theta - \pi), \\ z = a(1 - \cos\theta), \end{cases} \quad (0 \le \theta \le 2\pi), \tag{1}$$

To be specific, $\theta$ is the included angle between the radius of point A and the $Z$-axis; $\theta$ is equal to 0 at the initial position, while $\theta$ is equal to $2\pi$ at the end position. In the XOZ coordinate system, the $Z$-axis is regarded as the tool axis. The tool radius is equal to $\pi a$; the height of the end tooth is the diameter $2a$ of the selected circle. As shown in Figure 1b, a space rectangular coordinate system $\delta_t(X_t, Y_t, Z_t)$ is established with the axis direction of the tool as the $Z$-axis. The revolving surface formed by the cycloidal profile revolving around $Z$-axis is the contour surface of the revolving cycloid milling cutter, and can be expressed as:

$$S_A = \{a(\theta - \sin\theta - \pi)\cos\varphi, a(\theta - \sin\theta - \pi)\sin\varphi, a(1 - \cos\theta)\}, \tag{2}$$

Among them, $\varphi$ and $\theta$ are surface parameters. Then, in the coordinate system $\delta_t$, the contour surface of the revolving cycloid milling cutter can be expressed as:

$$\begin{cases} S_0 = \{a(\theta - \sin\theta - \pi)\cos\varphi_1, a(\theta - \sin\theta - \pi)\sin\varphi_1, a(1 - \cos\theta)\}, \\ S_1 = \{\pi a \cos\varphi_2, \pi a \sin\varphi_2, u\} \ (u > 0), \end{cases} \tag{3}$$

where, $S_0$ and $S_1$ are the revolving cycloid contour surface and the cylinder contour surface, respectively.

The orthogonal helix cutting edge is applied in the revolving cycloid milling cutter, while the intersection line between the orthogonal helix surface and the tool rotating surface is used as a cutting edge curve. The orthogonal helicoid can be expressed as:

$$S_B = \left\{ R_S \cos\varphi, R_S \sin\varphi, \frac{P_z \varphi}{2\pi} \right\} (0 \le R_S \le R), \tag{4}$$

Regarding the orthogonal helix cutting edge, the helical pitch $P_Z$ should satisfy the following equation:

$$\frac{P_z}{2\pi R} = \frac{1}{\tan\beta}, \tag{5}$$

Combining Equations (3)–(5), upon calculation, $\varphi$ can be expressed as:

$$\varphi = \frac{\tan\beta_0(1 - \cos\theta)}{\pi}, \tag{6}$$

where $\beta_0$ is the helix angle of the cutting edge of the cylindrical edge of the revolving cycloid milling cutter. Substituting Equation (6) into Equation (3), one can obtain a mathematical model for the orthogonal helix cutting edge curve of the revolving cycloid milling cutter, as shown below:

$$r_1 = \begin{cases} \{a(\theta - \sin\theta - \pi)\cos\varphi_1, a(\theta - \sin\theta - \pi)\sin\varphi_1, a(1 - \cos\theta)\}, \\ \varphi_1 = \frac{\tan\beta_0(1-\cos\theta)}{\pi}, \end{cases} \tag{7}$$

The mathematical model of the cutting edge curve of the cylindrical edge of the tool is:

$$r_2 = \begin{cases} \{R \sin\varphi_2, R \cos\varphi_2, u\}, \\ \varphi_2 = \frac{\tan\beta_0}{R} u, \end{cases} \quad (u > 0). \tag{8}$$

When a two-tooth revolving cycloid milling cutter is taken for an example, the contour surface of the revolving cycloid milling cutter and the orthogonal helix cutting edge curve model are simulated numerically on the basis of the Matlab software. The simulation result of the orthogonal helix cutting edge curve of the revolving cycloid milling cutter is shown in Figure 2. The radius of the revolving cycloid milling cutter is 5 mm, while curve 1, curve 2 and curve 3 are the orthogonal helix cutting edge curves when the helix angle $\beta$ is 50°, 40° and 30°, respectively.

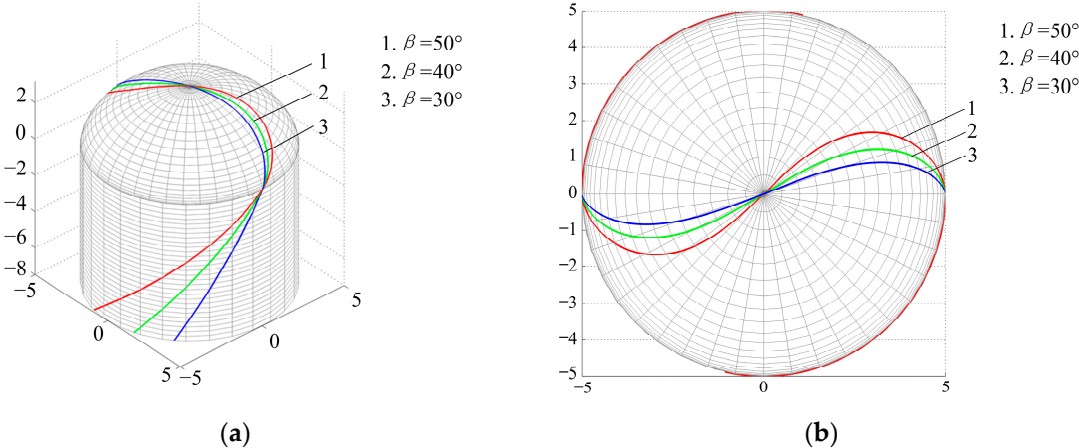

(**a**)  (**b**)

**Figure 2.** Cutting edge curve simulation of revolving cycloid milling cutter for different helical angles: (**a**) The cutting edge curve simulation for a cylindrical edge; (**b**) the cutting edge curve simulation for end edge.

### *2.2. Comparison of Cutting States of the Revolving Cycloid Milling Cutter and the Ball-end Milling Cutter*

(1)  Comparison of equivalent helix angles at the tool-workpiece cutting zone

Equivalent cutting helix angles of the revolving cycloid milling cutter and the ball-end milling cutter with the same radius are compared in Figure 3. In Figure 3, $\psi$ presents the machining inclination angle, while point $B_1$ and point $B_2$ are the tool-workpiece contact points of the revolving cycloid milling cutter and the ball-end milling cutter, respectively.

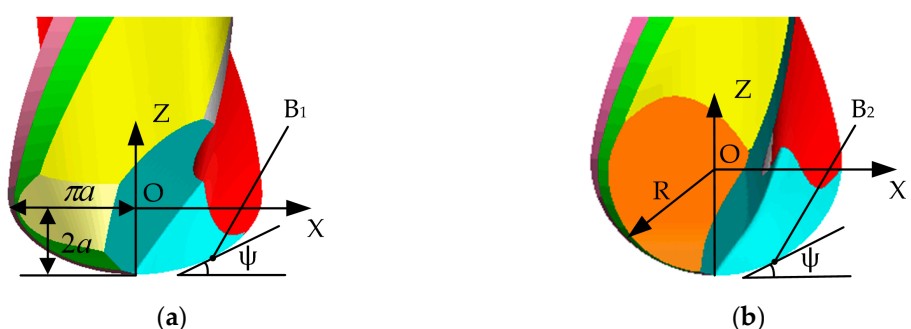

(**a**)  (**b**)

**Figure 3.** Comparison of equivalent helix angle for ball-end milling cutter and revolving cycloid milling cutter: (**a**) Revolving cycloid milling cutter; (**b**) ball-end milling cutter.

Based on the orthogonal helix edge curve, its helix angle can be generally expressed as:

$$\beta = \arccos\left(\frac{ds \cdot \partial s}{|ds| \cdot |\partial s|}\right), \tag{9}$$

Then, the helix angle $\beta_1$ of the cutting edge curve of the revolving cycloid milling cutter at point $B_1$ can be expressed as:

$$\beta_1 = \arccos \sqrt{\frac{2\pi^2(1 - \cos\theta)}{2\pi^2(1 - \cos\theta) + (\theta - \sin\theta + \pi)^2 \tan^2\beta_0 \sin^2\theta}}, \tag{10}$$

of which, the relationship of $\theta$ and the machining inclination angle $\psi$ satisfies the following equation:

$$\theta = \arccos\left(\frac{\tan^2\psi - 1}{\tan^2\psi + 1}\right), \tag{11}$$

The helix angle $\beta_2$ of the cutting edge curve of the ball-end milling cutter at point $B_2$ can be expressed as:

$$\beta_2 = \arccos \sqrt{\frac{R^4}{R^4 + (R^2 - R^2\cos^2\psi)^2 \tan^2\beta_0}}, \tag{12}$$

With the machining inclination angle $\psi$ as a parameter, $\beta_1$ and $\beta_2$ are simulated numerically. The result is shown in Figure 4.

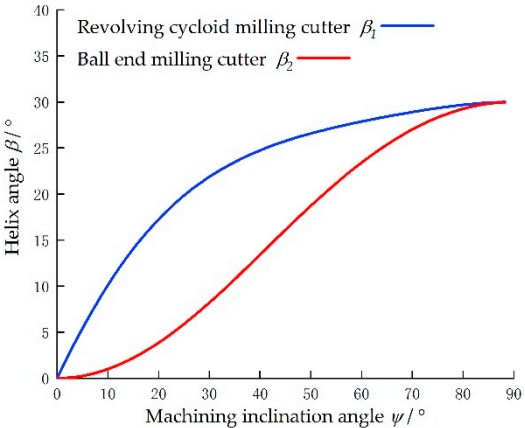

**Figure 4.** Comparison of the equivalent cutting helix angle under different machining inclination angle.

According to the simulation and cutting experiment results that have been completed, the larger helix angle can improve the cutting process. As can be seen from the figure, $\beta_1$ is greater than $\beta_2$ in the process of the machining inclination angle changing from 0° to 90°. Specifically, when the machining inclination angle is greater than 5° and less than 60°, an apparent advantage can be found in $\beta_1$, indicating that the actual cutting helix angle of revolving cycloid milling cutter is larger than that of the ball-end milling cutter, the cutting edge is longer, and the cutting process will be more stable.

(2) Comparison of scallop heights of the workpiece surface

The scallop heights of the revolving cycloid milling cutter and the ball-end milling cutter with the same radius are compared in Figure 5. It should be noted that $h_1$ and $h_2$ are machining scallop heights of the revolving cycloid milling cutter and the ball-end milling cutter. $r$ is the distance between the tool center point $O_1$ of the revolving cycloid milling cutter and the highest point of the scallop height $h_1$.

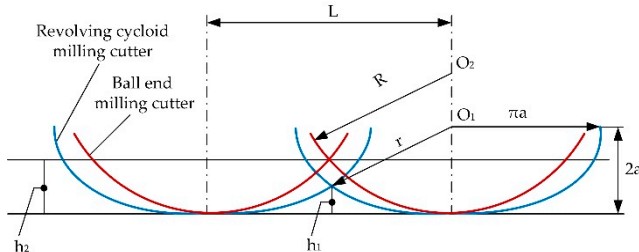

**Figure 5.** Comparison of scallop height of two kinds of milling cutters.

The scallop height $h_1$ of the revolving cycloid milling cutter is:

$$h_1 = 2a - \frac{\sqrt{4r^2 - L^2}}{2},$$ (13)

of which, $= \sqrt{[a(\theta - \sin\theta + \pi)]^2 + [a(\cos\theta - 1)]^2}$.

The scallop height $h_2$ of the ball-end milling cutter is:

$$h_2 = R - \frac{\sqrt{4R^2 - L^2}}{2},$$ (14)

When the tool radius is 3 mm, 6 mm and 10 mm, respectively, the corresponding relationship between the scallop height $h$ and the path interval $L$ during the machining of the revolving cycloid milling cutter and the ball-end milling cutter is shown in Figure 6. By analyzing the figure, the value of the scallop height of the workpiece surface of the revolving cycloid milling cutter is relatively small when the path interval is consistent, showing that the machining efficiency of the revolving cycloid milling cutter is significantly improved with the comparison of the ball-end milling cutter.

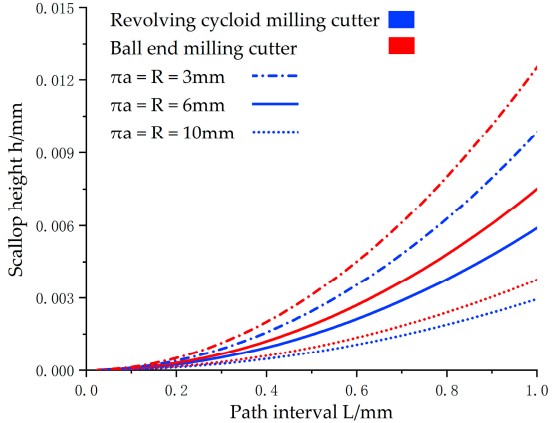

**Figure 6.** Scallop height under different path intervals.

## 3. Fabrications and Detection of the Revolving Cycloid Milling Cutter

### 3.1. Establishment of the Grinding Model for the Rake Face of the Revolving Cycloid Milling Cutter

In order to establish a mathematical model for the grinding trajectory of the rake face of the revolving cycloid milling cutter, the tangential vector and the normal vector along the end surface of the cutting edge are considered as basic components of the cutting edge. On this basis, the bottom boundary curve of the end tooth rake face can be obtained with the rake angle of the tool and the depth parameter of the chip hold groove of the end tooth, so as to derive the position and angle of grinding wheels when the rake face of the end tooth is grinding.

As shown in Figure 7, the tangent vector at any point $S_j$ on the cutting edge is $\vec{Q}_j$; the normal vector is $\vec{F}_j$; the cross product of $\vec{Q}_j$ and $\vec{F}_j$ is the vector $\vec{B}_j$. Vectors $\vec{F}_j$, $\vec{Q}_j$, and $\vec{B}_j$ can be expressed with the cutting edge curve $r$:

$$\begin{cases} \vec{Q}_j = \frac{\dot{r}}{\|\dot{r}\|} = [Q_{xj}, Q_{yj}, Q_{zj}], \\ \vec{F}_j = \frac{r}{\|r\|} = [F_{xj}, F_{yj}, F_{zj}], \\ \vec{B}_j = \vec{Q}_j \times \vec{F}_j = [B_{xj}, B_{yj}, B_{zj}], \end{cases} \tag{15}$$

To be specific, the normal forward angle $\gamma$ is measured through the normal surface $P_n$. The rake face can be deemed as an extending surface of $S_jV_j$ on the normal surface. The grinding track curve can be determined by determining the functional relationship between parameter $z$ and radial depth $h(z)$. Parameter $z$ is evaluated from 0 to 2a of the Z-axis in Figure 1a, and the relationship between parameter $z$ and parameter $\theta$ is shown in Equation (1).

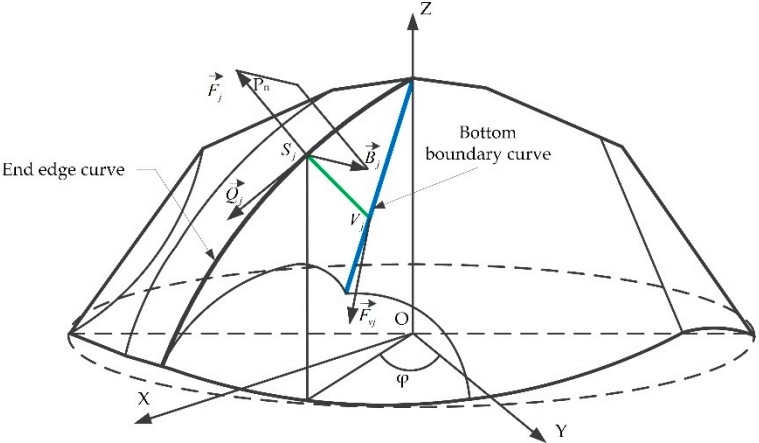

**Figure 7.** Mathematical model of the rake face for revolving cycloid milling cutter.

By taking the curve in Figure 8 for example, the functional relationship $h(z)$ is established, which is a parabola. Its definition domain and value domain are as follows: $z \in (-A, 0); h(z) \in (0, h_{max})$. The vertex of the curve is $(-A, 0)$. The curve passes point $(0, h_{max})$. Then $h(z)$ can be expressed as:

$$h(z) = h_{max}\left(1 + \frac{z}{A}\right)^2, \tag{16}$$

The bottom boundary curve equation is determined by the depth function $h(z)$ of the rake face, while the position and the angle of the grinding wheel during grinding are determined by the bottom boundary curve equation. The position vector $r_{ovj}$ between point $V_j$ on the bottom boundary curve and point $S_j$ on the cutting edge is:

$$r_{ovj} = r_{osj} + S_jV_j = [x_{vj}, y_{vj}, z_{vj}], \tag{17}$$

where, $S_jV_j = -h(u)cos\gamma\vec{F}_j - h(u)sin\gamma\vec{B}_j$. The bottom boundary curve is parameterized:

$$r_{ovj} = r_{ov}(\varphi) = [x_v(\varphi), y_v(\varphi), z_v(\varphi)], \tag{18}$$

Then, the normal vector $\vec{F}_{vj}$ of a point $V_j$ on the bottom boundary curve of the rake face of the end tooth is:

$$\vec{F}_{vj} = \frac{dr_{ovj}^2/d\varphi^2}{\|dr_{ovj}^2/d\varphi^2\|}, \tag{19}$$

Thereby, the parametric equation of the rake face of the end tooth is:

$$S_R(\varphi, m) = (1-m)r_{osj} + mr_{ovj} = r_{osj} - mh(u)\left(\cos\gamma\vec{F}_j + \sin\gamma\vec{B}_j\right), \tag{20}$$

where the coefficient $m$ is between 0 and 1. In order to avoid interference between the grinding wheel and the bottom boundary curve, when the vector $\vec{F}_{vj}$ enters the grinding wheel side plane at point $V_j$, the projection on the principal normal vector of the bottom boundary curve must pass through the center of the grinding wheel side. The normal vector $\vec{F}_{vj}$ of the bottom boundary curve intersects with the axis of the grinding wheel. When the conditions listed above are satisfied, the position and angle of the grinding wheel can be derived from the established mathematical model. The grinding model of the flank face of the revolving cycloid milling cutter and the position and angle of grinding wheels can be also deduced with the above method.

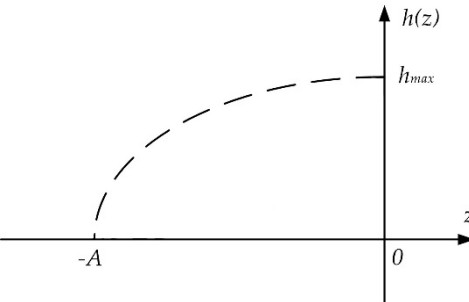

**Figure 8.** The functional relation between the parameter $z$ and $h(z)$.

### 3.2. Study on the Fabrication and Testing of the Revolving Cycloid Milling Cutter

For ensuring the grinding precision of the revolving cycloid milling cutter, based on industrial experience, the manufacturing accuracy specifications of cutting tools are selected, its indexes are requested as follows: Edge diameter should be ±0.01 mm; the length of the end tooth should be ±0.005 mm with the helix angle of ±0.05°, the rake angle of ±0.05°, and the clearance angle of ±0.05°; the tool diameter should be 10 mm with the helix angle of 30°, the rake angle of 6°, and the clearance angle of 8°. The Saacke five-axis tool grinding center and the Zoller G3 tool test center are applied in grinding of the revolving cycloid milling cutter and in the tool geometric measurement, respectively. The grinding field and the testing results are shown in Figure 9.

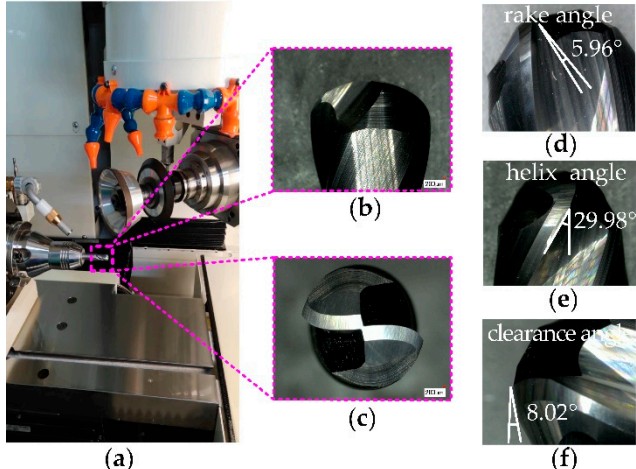

**Figure 9.** Grinding and geometrical angle detection of revolving cycloid milling cutter: (**a**) Fabrication setup of revolving cycloid milling cutter; (**b**,**c**) front view and top view of revolving cycloid milling cutter; (**d**–**f**) geometry angle detection results.

Figure 10a is the field for the tool test center testing the machining precision of the revolving cycloid milling cutter. Measurement results are shown in Figure 10b.

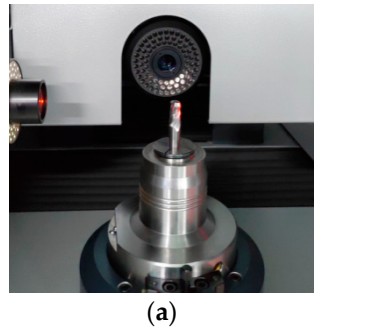

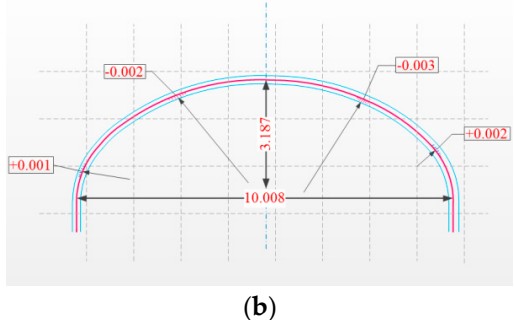

**Figure 10.** Profile accuracy detection of a revolving cycloid milling cutter: (**a**) Test equipment; (**b**) test results.

The red line in the figure presents the theoretical contour, while the blue line presents the limit deviation range of grinding precision. According to testing results of grinding precision, the edge diameter, the height of the end tooth, and the grinding precision of cycloid cutting edge can satisfy the precision requirements of the limit deviation range. Margins within the limit deviation range can be found at large curvatures on both sides of the contour, while overcut within the limit deviation range can be found at small curvatures. It is mainly caused by the grinding and feeding speed that failed to adjust according to the change of the removal rate of materials. The grinding and feeding speed will be optimized to enhance the grinding precision in a local area of the tool.

## 4. Study on the Cutting Performance of the Revolving Cycloid Milling Cutter

### 4.1. Test Conditions

The titanium alloy labeled as TC11 was selected as the cutting material for the test. Meanwhile, the revolving cycloid milling cutter and the ball-end milling cutter with consistent geometric parameters were used for the comparative test. As shown in Figure 11, the rectangular workpiece was clamped at the inclination angle of 20° to have down milling and cutting along the angle. The effective cutting speed was set as 100 m/min; the feed rate per tooth was 0.06 mm/z; the axial cutting depth was 0.2 mm; while the path interval was 0.2 mm. A Kister9171A revolving dynamometer was adopted in the

test process to collect the cutting force. When the cutting length was up to 25 m, a VHX-1000 3D microscope system with a super depth of field was adopted to observe wear topography and wear amount of the rake face and the flank face of the tool. Meanwhile, the workpiece surface at this stage was kept, and the surface roughness of the workpiece was observed using a Taylor Surface CCI white light interferometer.

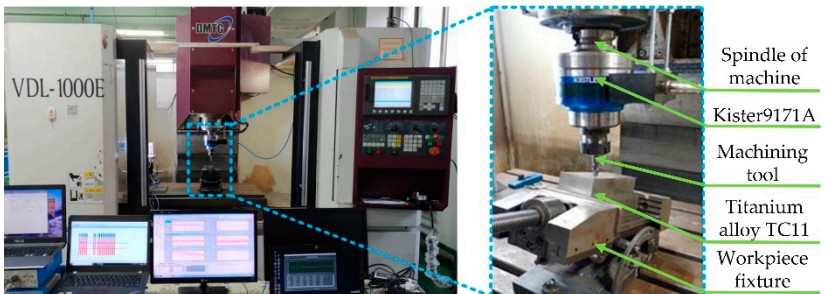

**Figure 11.** Cutting performance test layout.

*4.2. Results and Discussion*

4.2.1. Effects of Tool Wear on Cutting Force

When TC11 materials were milled by the two milling cutters under consistent cutting parameters, the maximum cutting forces of all directions are compared in Figure 12. It can be seen from the figure that the changing magnitudes of the tangential force and the axial force of the two milling cutters were remarkably greater than that of the radial force. The ratio of the axial force to the tangential force was 0.63, 0.53, 0.67, 0.57 and 0.71, 0.60, respectively, at the milling lengths of 75 m, 150 m and 225 m. The smaller ratio of the axial force to the tangential force of the revolving cycloid milling cutter shows a more stable cutting process. With the increase of tool wear, the changing trends of cutting forces in three directions were basically the same. The increase of cutting force of revolving cycloid milling cutter is small. The main reason is that the effective cutting edge length of revolving cycloid milling cutter was obviously longer than that of ball-end milling cutter, which produced smaller chip thickness and friction resistance in processing. In particular, when the milling length was 225 m, the tool wear of ball-end milling cutter was more serious, and the increase of cutting force in all directions was more obvious.

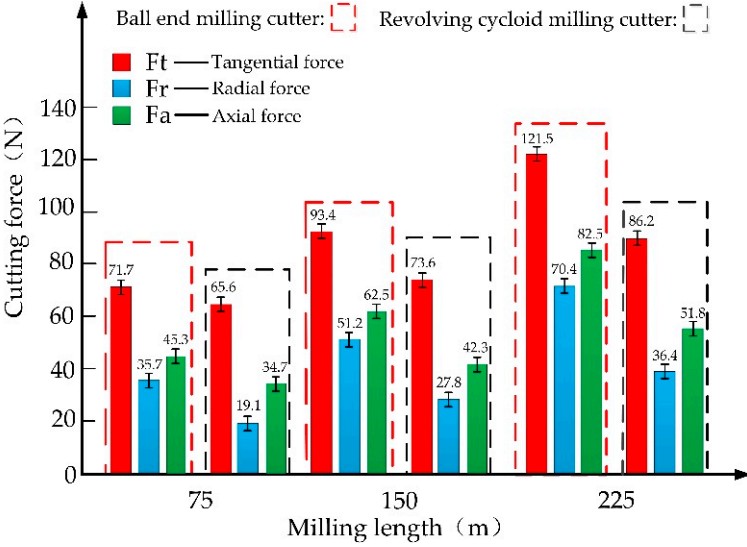

**Figure 12.** Comparison of the maximum cutting force between two kinds of cutters.

#### 4.2.2. Tool Wear

Figure 13 shows the tool wear morphology of ball-end milling cutters with different milling lengths under the same milling parameters. It can be seen from the figure that when the milling length reached 75 m, there were obvious scratch marks on the rake face, and a certain width of wear band was formed on the flank face. The wear of the tool sped up with the progress of cutting. When the milling length was 150 m, chipping and notches generated under the effect of impact load were witnessed in the rake face of the ball-end milling cutter. Meanwhile, the width of the wear land in the flank face was extended greatly along the direction away from the cutting edge. When the milling length was 225 m, an increasing number of dense notches could be found in the rake face of the ball-end milling cutter, while a spoon-shaped wear zones was formed in the flank face.

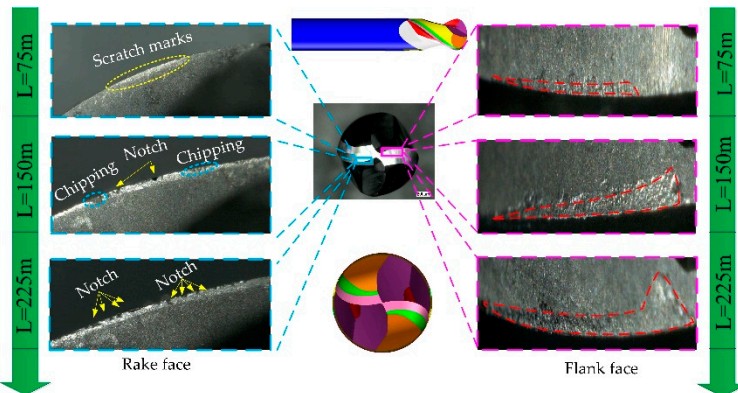

**Figure 13.** Tool wear morphology of ball-end milling cutter with different milling length.

The tool wear morphology of the revolving cycloid milling cutter under a consistent cutting condition is shown in Figure 14. When the milling length reached 75 m, the micro-chipping phenomenon of revolving cycloid milling cutters appeared. However, as the cutting process advanced, no obvious tool material chipping could be witnessed in the rake face of the revolving cycloid milling cutter with only a handful of notches when the milling length was 150 m. As the tool wore, there was no such a great amount of dense notches shown in the ball-end milling cutter when the milling length was 225 m. The wear land length of the flank face of revolving cycloid milling cutter was obviously longer than that of ball-end milling cutter, and the wear zones changed more uniformly which showed a slender and long shape. This is mainly because the revolving cycloid milling cutter had a larger cutting helix angle and longer cutting contact edge. It can effectively reduce the mechanical load and make the cutting process more stable, thus prolonging the tool life.

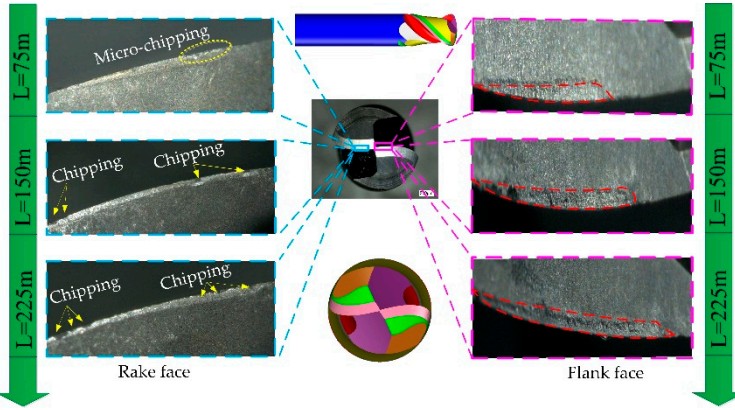

**Figure 14.** Tool wear morphology of the revolving cycloid milling cutter with different milling lengths.

### 4.2.3. Effects of Tool Wear on Machining Surface Topography

Figure 15 presents variations in the surface roughness of two milling cutters at different milling lengths under consistent milling parameters. As can be observed from the figure, the surface roughness first decreased and then increased with the increase of the milling length. By contrast, the surface roughness of the ball-end milling cutter increased more intensely, while the machining surface roughness of the revolving cycloid milling cutter increased slowly during the whole machining process.

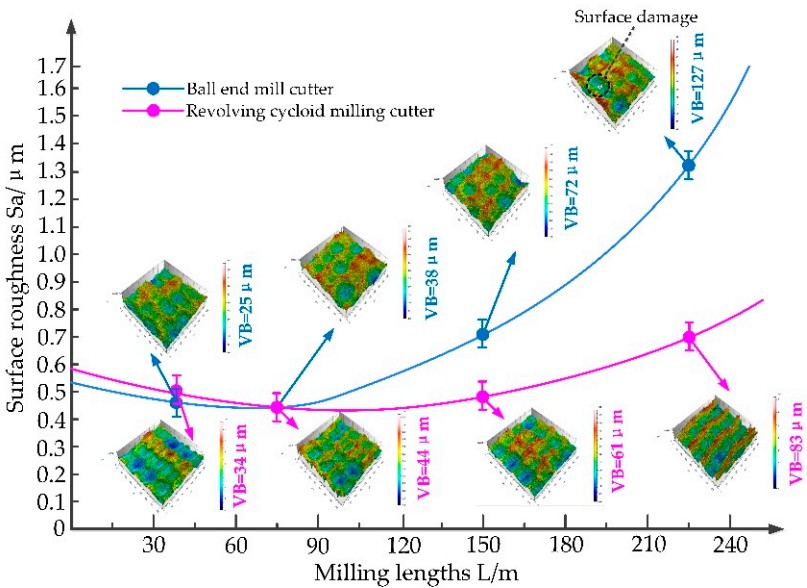

**Figure 15.** Variation of the 3D surface roughness with milling length.

When the milling length was within 75 m, there was no obvious difference between the machining surface roughness of the two tools. This may be due to measurement and manufacturing errors. When the milling length reached 150 m, the surface roughness of the revolving cycloid milling cutter was less than that of the ball-end milling cutter, and the machined surface was more regular and smoother. When the milling length was 225 m, the friction and extrusion of the flank face of the tool and the workpiece intensified by the spoon-shaped wear land caused by the ball-end milling cutter as the tool wear sped up. Consequently, the dramatically-dropped quality of the machining surface resulted from defects such as tearing and pit in the workpiece material. Nevertheless, the machining surface quality of the revolving cycloid milling cutter was more ideal, since it had a large equivalent cutting helix angle and a long effective cutting edge.

### 5. Conclusions

1. Based on the established mathematical model of the contour surface of the revolving cycloid milling cutter, a parametric equation of the orthogonal helix cutting edge has been proposed to numerically simulate the established contour and edge line model. Advantages of the revolving cycloid milling cutter over the equivalent cutting helix angle and the scallop height of the workpiece surface have been proved by simulated results.
2. The bottom boundary curve of the rake face was introduced. Based on the cutting edge equation and coordinate transformation, the five-axis grinding trajectory equation of the revolving cycloid milling cutter for rake face has been derived. The studies on grinding and detection of revolving cycloid milling cutter were carried out, and the quantitative evaluation of the grinding accuracy of the developed cutter was realized.
3. Tests on the cutting force of the cutting titanium alloy TC11 of the revolving cycloid milling cutter and the ball-end milling cutter were studied comparatively. It can be found that the revolving

cycloid milling cutter can significantly lower the axial force, the tangential force and the ratio of the axial force to the tangential force. Even under tool wear conditions, its cutting forces in all directions were increased slowly, showing excellent cutting stability.

4.　Compared with the ball-end milling cutter, the wear land of the revolving cycloid milling cutter in the cutting titanium alloy TC11 was narrow and long. As the wear sped up, the spoon-shaped wear gathering zone found in the ball-end milling cutter did not happen to the revolving cycloid milling cutter. The machining quality of the revolving cycloid milling cutter was relatively ideal.

**Author Contributions:** Conceptualization, G.W. and T.C.; methodology, X.L.; software, W.G.; validation, G.W., W.G. and B.Y.; formal analysis, T.C.; investigation, W.G.; resources, B.Y.; data curation, T.C.; writing-original draft preparation, G.W.; writing-review and editing, W.G.; visualization, T.C.; supervision, X.L.; project administration, T.C.; funding acquisition, X.L.

**Funding:** This research was funded by National Natural Science Foundation's international (regional) cooperation and exchange project: The basis and application of intelligent cutting technology based on open CNC system (51720105009).

**Conflicts of Interest:** The authors declare no conflict of interest.

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
