# Peer review of "Study on the Design and Cutting Performance of a Revolving Cycloid Milling Cutter"

_applsci, doi:10.3390/app9142915_

Round 1
Reviewer 1 Report
Study on design and cutting performance of 2 revolving cycloid milling cutter
Review
This article is really interesting and very complete. It includes most of aspects from tool design, tool grinding and tool testing. That is very great and unusal.
Comments
The geometrical approach for tool definition is interesting. The authors do not mention much other work on different approached for geometrical tool definition and analysis. It would mention among
Guillaume FROMENTIN, Gérard POULACHON - Geometrical analysis of thread milling – Part 1: Evaluation of tool angles - The International Journal of Advanced Manufacturing Technology - Vol. 49, n°1, p.73-80 – 2010 https://sam.ensam.eu/bitstream/handle/10985/7470/LABOMAP_IJAMT_FROMENTIN_2010_2.pdf?sequence=1&isAllowed=y
Equation 5: length unit on left side, not on right side. Any problem.
Alpha: machining inclination angle. It makes some trouble. Usually alpha is rake or clearance angle. Inclination angle is the obliquity angle (thus it is also the helix angle for a cylindrical milling cutter). What you name inclination angle is named tool cutting edge angle according to ISO 3002 standard.
The definition of radial depth is not clear. Add to a figure please. Is Fig 8 with h(z) coherent with Fig 1(a).
There is not much explanation about geometrical analysis/formulation for ball end mill cutter.
It is written “cutting width”. Once again it may be not the proper name. the cut width is the length of the engaged cutting edge. In your case it may be the radial depth of cut.
Fig 12: Which force are they? The means ones? The max, the PtP?
The comparison and the analysis of cutting force is not much developed. If tool engagement and feed are identical (ap, ae, fz) for both cutter, then material removal rate is the same.
When modified tool profile it affects cut thickness. If global engagement remains constant (ae,ap) , specific cutting force increases when reducing cut thickness. Global cutting force would increase, unless, the helix angle is changing and share the cut thickness during the tool revolution. Furthermore, cutting geometry may slightly be affected by rake angle (1 to 2% / °).
Usually reducing tool cutting edge angle is the way to design cutter for higher feed rate.
If wear is slight lower with new cutter is may be only linked to smaller cut thickness.
About roughness, without wear the one with cycloid tool has to be better than the one with ball end milling cutter. It is the opposite. What about geometrical model value of scallop height for both cutters.
You used Sa and not Ra. Why? Sa may be affected by synchronisation of tool path.
About roughness, the use of your new cutter may be not obvious for complex surface machining. The tool contact point may be easily identified. CNC include 3D radius compensation for ball end mill not for this kind of cutter.
Author Response
1. The geometrical approach for tool definition is interesting. The authors do not mention much other work on different approached for geometrical tool definition and analysis. It would mention among.
Guillaume FROMENTIN, Gérard POULACHON - Geometrical analysis of thread milling – Part 1: Evaluation of tool angles - The International Journal of Advanced Manufacturing Technology https://sam.ensam.eu/bitstream/handle/10985/7470/LABOMAP_IJAMT_FROMENTIN_2010_2.pdf?sequence=1&isAllowed=y
Response: This article has been carefully studied and referenced. And the modified part is marked with red.
2.Equation 5: length unit on left side, not on right side. Any problem.
Response: Equation 5 has been modified. And the modified part is marked with red.
3. Alpha: machining inclination angle. It makes some trouble. Usually alpha is rake or clearance angle. Inclination angle is the obliquity angle (thus it is also the helix angle for a cylindrical milling cutter). What you name inclination angle is named tool cutting edge angle according to ISO 3002 standard.
Response: To avoid ambiguity, the letter A has been used to indicate the machining inclination angle. And the modified part is marked with red.
4. The definition of radial depth is not clear. Add to a figure please. Is Fig 8 with h(z) coherent with Fig 1(a).
Response: Parameter z is evaluated from 0 to 2a of the Z-axis in Figure 1(a). And the modified part is marked with red.
5. There is not much explanation about geometrical analysis/formulation for ball end mill cutter.
Response: The content has been added to the article and marked in red. The red curve in Figure 4 describes the change rule of the effective cutting helix angle of ball end milling cutter with the machining inclination angle.
6. It is written “cutting width”. Once again it may be not the proper name. the cut width is the length of the engaged cutting edge. In your case it may be the radial depth of cut.
Response: “cutting width” has been amended as “path interval”. And the modified part is marked with red.
7. Fig 12: Which force are they? The means ones? The max, the PtP?
Response: Fig. 12 has been modified by replacing Fx, Fy and Fz in the figure with Ft(tangential force), Fr(radial force) and Fa(axial force), which respectively represent the maximum cutting force in the three directions. And the modified part is marked with red.
8. The comparison and the analysis of cutting force is not much developed. If tool engagement and feed are identical (ap, ae, fz) for both cutter, then material removal rate is the same.When modified tool profile it affects cut thickness. If global engagement remains constant (ae,ap) , specific cutting force increases when reducing cut thickness. Global cutting force would increase, unless, the helix angle is changing and share the cut thickness during the tool revolution. Furthermore, cutting geometry may slightly be affected by rake angle (1 to 2% / °).
Response: Fig 12 has been modified to add the change of cutting force of two kinds of cutting tools when the milling distance is 150 meters in the figure, showing in more detail the change rule and similarities and differences of cutting force of two kinds of tools in the milling process. The modified parts have been marked in red.
9. Usually reducing tool cutting edge angle is the way to design cutter for higher feed rate.If wear is slight lower with new cutter is may be only linked to smaller cut thickness.About roughness, without wear the one with cycloid tool has to be better than the one with ball end milling cutter. It is the opposite. What about geometrical model value of scallop height for both cutters.
Response: The difference of machining surface roughness between the two cutting tools is not obvious at the beginning of cutting .This may be due to errors in tool manufacturing and experimental measurements. The modified parts have been marked in red in 4.2.3.
10. You used Sa and not Ra. Why? Sa may be affected by synchronisation of tool path.
Response: In this paper, 3D surface roughness is adopted to describe machining quality mainly because Sa can more accurately reflect surface microstructure and state. For example, the ball end milling cutter shown in Figure 15 of this paper appears damage on the surface at the later processing stage, but the two-dimensional surface roughness Ra cannot describe this phenomenon.
Reviewer 2 Report
Article title: "Study on design and cutting performance of revolving cycloid milling cutter" is prepared correctly.
Suitable for publication in the Journal Applied Sciences.
Author Response
Suitable for publication in the Journal Applied Sciences.
Response: This article has been carefully studied and referenced. And the modified part is marked with red.
Reviewer 3 Report
The manuscript is interesting and well prepared. However, I have some suggestions:1. In abstract, lines 10-14 are obvious and unnecessary.
2. In Figure 12, Fx, Fy, Fz must by replaced through Ft, Fr and Fa.
3. In the manuscript only the maximum cutting speed is specifed. Please specife also the effective cutting speed for both tools.
4. Can you estimate the error in the forces and surface roughness measurements? In figures 12 and 15 error bars are missing.
5. The surface roughness measurement procedure could be better explained. Please specify the measurements and analysis parametrers.
Author Response
1. In abstract, lines 10-14 are obvious and unnecessary.
Response: Lines 10-14 have been overwritten. The corresponding parts have been marked in red.
2. In Figure 12, Fx, Fy, Fz must by replaced through Ft, Fr and Fa.
Response: Fx, Fy and Fz in Figure 12 have been replaced with Ft, Fr and Fa respectively, and marked in red.
3. In the manuscript only the maximum cutting speed is specifed. Please specife also the effective cutting speed for both tools.
Response: In this paper, the effective speeds of both tools are 100m/min. The modified part is marked in red in the 4.1 experimental design part of this paper.
4. Can you estimate the error in the forces and surface roughness measurements? In figures 12 and 15 error bars are missing.
Response: Error bars have been added to Figure 12 and Figure 15 and marked in red.
5. The surface roughness measurement procedure could be better explained. Please specify the measurements and analysis parametrers.
Response: In this paper, the three-dimensional morphology of the machined surface is measured by Taylor Surface CCI white light interferometer, and the height parameter Sa is selected to characterize the three-dimensional morphology of the extracted surface. The model of the measuring instrument has been given in the experimental design section 4.1 of this paper, and it has been marked in red.
Reviewer 4 Report
The paper addresses a very interesting topic, both for the academia than for the industries.
The overall organization of the paper is good and the flow of information and the reasoning is straight. Generally speaking, I suggest to review the English and to avoid non “formal” statements like “The above researches have done a lot of work”.
The detailed improvements that could be carried out are the following:
- These two statements looks conflicting, please explain better in which specific field there are not many relevant scientific papers, I presume on cycloid tools, and I do not think the only reason is the lack of processing equipment… “The above researches have done a lot of work in the design and manufacturing of new cutting tools. The existing researches focus on helical edge design, geometric angle optimization and grinding manufacturing of ball end milling cutter [14-20]. Due to the limitation of processing equipment and grinding method, there is little research on new cutting tools for titanium alloy processing”
- In paragraph 2, before explaining how a cycloid cutter is designed, it will be important to highlight why a cycloid is selected as best shape for a cutter.
- The description of the geometry of the cycloid cutter and its geometrical advantages respect to ball end cutter is clearly depicted. In order to improve further this part, please provide a motivation for this statement “?1, indicating that the actual cutting helix angle of revolving cycloid milling cutter is larger than that of the ball end milling cutter, the cutting edge is longer, and the cutting process will be more stable.”
- How are selected the specs for the manufacturing accuracy reported in paragraph 3? Based on industrial experience? Thanks to a sensitivity analysis? “its indexes are requested as follows: Edge diameter should be ±0.01mm; the length of the end tooth should be ±0.005mm with the helix angle of ±0.05°, the rake angle of ±0.05°, and the clearance angle of ±0.05°; the tool diameter should be 10 mm with the helix angle of 30°, the rake angle of 6°, and the clearance angle of 8°”
- No need to repeat the geometry of the tool in paragraph 4 “The diameter is 10mm with the helix angle of 30¬∞, the rake angle of 6¬∞, and the clearance angle of 8¬∞.”
- It would be interesting to see the evolution of cutting forces during the process and not only considering two points (Fig 12 - 75 and 225 m). Maybe showing the trend of only one component or the resultant vector. Use of an approach similar to the one used for the surface roughness will be appropriate.
Author Response
1. The overall organization of the paper is good and the flow of information and the reasoning is straight. Generally speaking, I suggest to review the English and to avoid non “formal” statements like “The above researches have done a lot of work”.
Response: This part of the text has been rewritten. And the modified part is marked with red.
2.The detailed improvements that could be carried out are the following:
-These two statements looks conflicting, please explain better in which specific field there are not many relevant scientific papers, I presume on cycloid tools, and I do not think the only reason is the lack of processing equipment… “The above researches have done a lot of work in the design and manufacturing of new cutting tools. The existing researches focus on helical edge design, geometric angle optimization and grinding manufacturing of ball end milling cutter [14-20]. Due to the limitation of processing equipment and grinding method, there is little research on new cutting tools for titanium alloy processing”
Response: Another reason may be the wide use of ball end milling cutters in the machining industry. And the modified part is marked with red.
2. In paragraph 2, before explaining how a cycloid cutter is designed, it will be important to highlight why a cycloid is selected as best shape for a cutter.
Response: The following statement has been added: “Cycloids are widely used in industry, and their width and height correspond to each other.”, and marked in red.
3. The description of the geometry of the cycloid cutter and its geometrical advantages respect to ball end cutter is clearly depicted. In order to improve further this part, please provide a motivation for this statement “?1, indicating that the actual cutting helix angle of revolving cycloid milling cutter is larger than that of the ball end milling cutter, the cutting edge is longer, and the cutting process will be more stable.”
Response: The following statement has been added: “According to the simulation and cutting experiment is taken, the larger helix angle can improve the cutting process.”, and marked in red.
3. How are selected the specs for the manufacturing accuracy reported in paragraph 3? Based on industrial experience? Thanks to a sensitivity analysis? “its indexes are requested as follows: Edge diameter should be ±0.01mm; the length of the end tooth should be ±0.005mm with the helix angle of ±0.05°, the rake angle of ±0.05°, and the clearance angle of ±0.05°; the tool diameter should be 10 mm with the helix angle of 30°, the rake angle of 6°, and the clearance angle of 8°”
Response: Based on industrial experience, the manufacturing accuracy specifications of cutting tools are selected in this paper. The added parts are marked in red in part 3.2 of this paper.
4. No need to repeat the geometry of the tool in paragraph 4 “The diameter is 10mm with the helix angle of 30°, the rake angle of 6°, and the clearance angle of 8°.”
Response: The corresponding part has been deleted.
5. It would be interesting to see the evolution of cutting forces during the process and not only considering two points (Fig 12 - 75 and 225 m). Maybe showing the trend of only one component or the resultant vector. Use of an approach similar to the one used for the surface roughness will be appropriate.
Response: Fig 12 has been modified to add the change of cutting force of two kinds of cutting tools when the milling distance is 150 meters in the figure, showing in more detail the change rule and similarities and differences of cutting force of two kinds of tools in the milling process. The modified parts have been marked in red.